# Targeting Ion Channels: Blockers Suppress Calcium Signals and Induce Cytotoxicity Across Medulloblastoma Cell Models

**DOI:** 10.3390/bioengineering12030268

**Published:** 2025-03-09

**Authors:** Darani Ashley Thammavongsa, Taylor N. Jackvony, Markus J. Bookland, Min D. Tang-Schomer

**Affiliations:** 1UConn Health, Department of Pediatrics, 263 Farmington Avenue, Farmington, CT 06030, USA; daraniashley@gmail.com (D.A.T.); taylornjackvony@gmail.com (T.N.J.); 2Connecticut Children’s Medical Center, 282 Washington St, Hartford, CT 06106, USA; mbookland@connecticutchildrens.org; 3The Jackson Laboratory for Genomic Medicine, 10 Discovery Drive, Farmington, CT 06030, USA

**Keywords:** ion channels, calcium signaling, potassium channels, cytotoxicity, Na/K ATPase inhibitors, calcium modulators, channel blockers, tumor cytotoxicity, targeted therapies, spontaneous calcium waves

## Abstract

Medulloblastoma (MB) groups 3 and 4 lack targeted therapies despite their dismal prognoses. Ion channels and pumps have been implicated in promoting MB metastasis and growth; however, their roles remain poorly understood. In this study, we repurposed FDA-approved channel blockers and modulators to investigate their potential anti-tumor effects in MB cell lines (DAOY and D283) and primary cell cultures derived from a patient with MB. For the first time, we report spontaneous calcium signaling in MB cells. Spontaneous calcium signals were significantly reduced by mibefradil (calcium channel blocker), paxilline (calcium-activated potassium channel blocker), and thioridazine (potassium channel blocker). These drugs induced dose-dependent cytotoxicity in both the DAOY and D283 cell lines, as well as in primary cell cultures of a patient with group 3 or 4 MB. In contrast, digoxin and ouabain, inhibitors of the Na/K pump, reduced the calcium signaling by over 90% in DAOY cells and induced approximately 90% cell death in DAOY cells and 80% cell death in D283 cells. However, these effects were significantly diminished in the cells derived from a patient with MB, highlighting the variability in drug sensitivity among MB models. These findings demonstrate that calcium signaling is critical for MB cell survival and that the targeted inhibition of calcium pathways suppresses tumor cell growth across multiple MB models.

## 1. Introduction

Medulloblastoma (MB) is a malignant, embryonic brain tumor that arises in the posterior fossa, most commonly within the cerebellum. The molecular profiling of MB has led to the identification of four distinct molecular subgroups: WNT, sonic hedgehog (SHH), group 3, and group 4 [1]. The WNT and SHH subgroups are characterized by constitutively active signaling pathways, which have been targeted with small-molecule inhibitors, resulting in relatively favorable prognoses [2,3]. In contrast, groups 3 and 4, which account for a significant proportion of MB cases, are poorly understood in terms of their genetic and molecular drivers, which has led to less effective treatment strategies. Therefore, the development of therapeutic treatments targeting novel subgroup-specific molecular and epigenetic factors are urgently needed [4].

Aberrant gene expression in group 3 and 4 MBs often involves receptor and ion channel proteins, such as metabolic glutamate receptors and potassium channels (KCNs) [5]. For example, the EAG2 potassium channel has been implicated in MB progression, with studies showing that its inhibition via the potassium channel blocker thioridazine reduces MB growth and metastasis in xenograft models [6]. EAG2 facilitates local cell volume dynamics that support cell motility, as demonstrated by its enrichment at the trailing edge of migrating MB cells. Additionally, a systems biology analysis of over 1300 FDA-approved drugs identified the cardiac glycoside family as potential candidates for targeting group 3 and 4 MBs, with the sodium-potassium pump inhibitor digoxin demonstrating significant cytotoxicity in MB cell lines and improving survival in orthotopic xenograft models [7]. These findings suggest that ion channels and pumps represent promising therapeutic targets for these MB subtypes.

Ion channels play critical roles in the function of cells and their survival that directly impact cancer progression [8]. Key mechanisms for cell cycle control have been found to rely on potassium [9,10] and calcium channels [11]. Cell proliferation and apoptosis involve alterations in calcium homeostasis and calcium signaling [12,13,14,15]. Ion channel functions are required for many characteristic features of the aggressive cancer cell behavior such as migration, invasion and metastasis as well as proliferation and therapy resistance. For example, as a tumor expands and invades the surrounding tissue, the cell volume change is mainly regulated by ions like potassium and chloride [16,17]. These ion channels manipulate the movement of water molecules across the cancer cell to propel the cells forward in confined spaces [18,19,20].

Ion channels are considered particularly “druggable” due to their accessibility on the cell surface and the availability of existing medications designed to target them, primarily for the treatment of cardiovascular and neurological disorders [21,22]. However, the diversity of ion channels necessitates a deeper understanding of the specific channels associated with MB and their functional roles. Given that ion channels critically regulate calcium homeostasis [23], we hypothesized that alterations in calcium signaling could serve as functional readouts of the ion channel activity.

In this study, we explored the potential of repurposing FDA-approved ion channel blockers as therapeutic agents for group 3 and 4 MB We selected inhibitors that target major ion channel classes, including thioridazine and paxilline (K^+^ channels), mibefradil (Ca^2+^ channels), and riluzole (Na^+^ and glutamate signaling). To evaluate their effects, we performed calcium imaging using the fluorescent indicator Fluo-4 [24] to monitor real-time calcium signaling changes in response to the drug treatments. Experiments were conducted on MB cell lines (DAOY [25] and D283 [26]) as well as the primary cell cultures of a patient with MB. While the MB cell lines were established decades ago, later molecular profiling classified DAOY as a SHH type and D283 as a group 3 or 4 type MB [27]. The primary MB cells used for the live imaging and drug studies were derived from group 3 or 4 MB (non-WNT, non-SHH, p53 wild-type). Our findings establish a mechanistic link between calcium signaling and MB cell viability and identify channel blockers—mibefradil, thioridazine, and paxilline—as potent inhibitors of MB cell growth in vitro. This study lays the groundwork for repurposing channel blockers as potential therapeutic agents for high-risk MB subtypes.

## 2. Materials and Methods

### 2.1. Medulloblastoma Tumor Tissue

Human patient brain tissue was obtained by tumor resection neurosurgery in Connecticut Children’s Medical Center (CCMC) at Hartford, Connecticut. The procedures were approved by the Institutional Review Boards of UConn Health Center and CCMC (IRB #13-035). Informed consent was obtained from all human patients prior to the surgery. All methods were performed in accordance with the guidelines and regulations of the approved IRB protocol. Tissue specimen was transported in chilled RPMI-1640 medium (Sigma-Aldrich, St Louis, MO, USA) containing 1% penicillin-streptomycin (Pen/Strep, Thermo Fisher, Waltham, MA, USA) and 5% fetal bovine serum (FBS) on an ice pack from the operation room to the laboratory in <4 h post-surgery.

### 2.2. Patient-Derived Primary Medulloblastoma Cell Culture

Primary tumor cell culture was established as described in our previous studies [28]. Briefly, the tissue specimen was weighed, minced with a sterile razor blade, and treated with a cocktail of enzymes (DNase I, 50 U; dispase II, 5 U; collagenase I, 1 U and collagenase IV, 10 mg/mL in 10 mL 0.5% trypsin-EDTA solution) at 37 °C for 20 min, followed by neutralization with a 10 mL trypsin inhibitor (0.5%, *w*/*v*) solution and gentle pipetting. The tissue dissociation solution was filtered with a 100 µm cell strainer (Fisher Scientific, Suwannee, GA, USA) and single cell suspension was collected. The primary cells were plated at 10,000 cells/well in 96-well plates. Culture medium used was 1:1 mixture of “NB” medium, e.g., NeuralBasal\B27 (Invitrogen, Grand Island, NY, USA) supplemented with 20 ng/mL recombinant human fibroblast growth factor, basic-154 (FGF, ConnStem, Cheshire, CT, USA), 20 ng/mL human epidermal growth factor (EGF, PeproTech, Rocky Hill, NJ, USA), and endothelial growth media EGM-2MV (Lonza, Walkersville, MD, USA). Upon cell confluency at 2–3 weeks post-seeding, the plates were subjected to drug treatment.

### 2.3. Cell Lines

DAOY and D283 cell lines (ATCC, Manassas, VA, USA) were cultured in Eagle’s minimum essential medium (MEM, ThermoFisher) supplemented with 10% fetal bovine serum and 1% streptomycin/penicillin. Cell cultures were maintained at 37 °C and in 5% CO_2_. At 80% confluence, the cultures were washed with PBS, detached with 0.25% trypsin, centrifuged, and subcultured at 5000 cells/well in 96-well plates. Upon cell confluency at 3–5 days, the plates were washed with PBS and incubated with serum-free media for 24 h before drug treatment.

### 2.4. Drug Treatment

All the drugs used in this study were purchased from the Alomone Labs (Jerusalem, Israel). Drug stock solution was prepared according to the manufacturer’s manual in the range of 0.1–10 mM. Serial dilution of the stock solution was made in serum-free MEM into 5–7 doses in triplicates. A total of 0.1% DMSO in MEM was used as vehicle control. At 48 and 96 h of drug treatment, the plates were retrieved for cell viability assay.

### 2.5. Cell Viability

LIVE/DEAD^®^ viability/cytotoxicity kit (Molecular Probes, Inc., Eugene, OR, USA) was used according to the manufacturer’s manual. Briefly, an equal volume (e.g., 100–150 µL) of 2X Calcein-AM (for live cells)- and ethidium homodimer-1 (EthD-1, for dead cells)-containing medium was added to each well and the plates were incubated for 20 min at 37 °C and 5% CO_2_. The stained plates were read at 528 nm for Calcein-AM and at 590 nm for EthD-1 with a microplate reader (Synergy™, BioTek Instruments, Winooski, VT, USA). Results were expressed as percentage of controls (0.1% DMSO-treated cells).

### 2.6. Calcium Imaging

Fluo-4 Direct^TM^ Calcium Assay kit (Molecular Probes, Inc., Eugene, OR, USA) was used according to the manufacturer’s manual. Briefly, equal volume (e.g., 100–150 µL) of 2X Flo-4 Direct^TM^ calcium reagent loading solution was added to wells containing cells in culture medium. After 30 min incubation at 37 °C, the plates were mounted onto an inverted microscope (Nikon TE2000, Kanagawa, Japan) within an environmental chamber with the temperature maintained at 37 °C and CO^2^ at 5%. Time-lapse fluorescence images were acquired at Ex/Em of 488/525 nm. To image the same well “before” and “after” drug treatment, baseline fluorescence time lapse images were collected first. Culture medium was subsequently replaced with drug-containing medium in situ, and “after”-treatment images were collected under the same optical setting.

NIH Image J software suite (version 1.53q) was used to quantify the fluorescence intensity. Circular selection was made for a cell body, and the fluorescence intensity of each frame composed the time-series for the cell. For each well, 10 cells were randomly selected. Custom-written R code in RStudio (version 4.3.0) was used to detect and quantify a calcium peak.

### 2.7. Flow Cytometry for Ion Channel Expression

Ion channel detection on MB cells was quantified by indirect flow cytometry. Cells were treated with 0.5% trypsin-EDTA (5 min) (Invitrogen) and 0.25% trypsin-EDTA (25 min) for 2D cultures and 3D tissue, respectively. Cell suspensions were mixed at 1:1 with medium containing 10% FBS and centrifuged at 300 g for 5 min. Cell pellets were re-suspended in PBS containing 2% FBS and stained on ice for 15 min with eFluor 780 (Affymetrix eBioscience, San Diego, CA, USA). Cells were washed in 2% FBS-containing PBS by centrifuging at 300 g for 5 min. For extracellular antibodies, cell pellets were re-suspended and stained with primary antibodies on ice for 30 min. For intracellular antibodies, cell pellets were fixed with 4% paraformaldehyde (Electron Microscopy Sciences, Hatfield, PA, USA) for 20 min, washed, permeabilized with PBS containing 0.1% Tween and 0.2% FBS for 20 min, and subsequently stained with primary antibodies on ice for 30 min. Cells stained with primary antibodies were washed, stained with goat anti-rabbit Alexa 488 secondary antibody (ThermoFisher) for 20 min, and washed again before proceeding to flow cytometry.

Primary ion channel antibodies were from Alomone Labs (Jerusalem, Israel): mGluR8 (rabbit polyclonal, extracellular, Cat# AGC-028), Kv1.1 (rabbit polyclonal, extracellular, Cat# APC-161), Kv1.5 (rabbit polyclonal, intracellular, Cat# APC-004), Kv2.1 (rabbit polyclonal, intracellular, Cat# APC-012), Kv2.2 (rabbit polyclonal, intracellular, Cat# APC-120), Kir2.1 (rabbit polyclonal, intracellular, Cat# APC-026), and sodium channels (Pan-NaV) (rabbit polyclonal, intracellular, Cat# ASC-003).

Flow cytometry was performed on a BD LSR II instrument equipped with 5 lasers and BD FACSDiva^TM^ software version 9.0 (BD Biosciences, San Jose, CA, USA). A total of 5000 cells per sample were counted and analyzed with FlowJo software version 10 (Flow-Jo, Ashland, OR, USA). Un-stained cells were used to set a gate for “live & single” cells. eFluor 780-stained cells were used to set the gate for “live” cells and “control” gates for each stain with a threshold of 0.5% (i.e., <0.5% cells were positive for the respective stain). Positive cell population corresponding to a stain was calculated using the same gate as that used for the control un-stained cell population.

### 2.8. Immunofluorescence Staining and Imaging

Cell cultures were fixed with 4% paraformaldehyde (Electron Microscopy Sciences) for 20 min, washed, and permeabilized with PBS containing 0.1% Triton X-100 (Fisher Scientific) and 4% normal goat serum (Jackson ImmunoResearch Labs, West Grove, PA, USA) for 20 min, followed by the incubation of primary antibodies overnight at 4 °C. After three 10 min PBS washes, cells were incubated with secondary antibodies for 1 h at room temperature, followed by extensive washes. The same primary ion channel antibodies for flow cytometry were used for immunostaining. Goat anti-rabbit Alexa 488 and 568 (1:250; Invitrogen) secondary antibodies were used. Fluorescence images were acquired on a Leica DM IL fluorescence microscope using excitation/emission (Ex/Em) of 470/525 nm for Alexa 488, and Ex/Em of 560/645 nm for Alexa 568.

### 2.9. RNA-Seq and Transcriptomic Profiling

RNAs were extracted with Qiagene AllPrep kit on a QiaCube automated station (Qiagen, Hilden, Germany). Samples were sequenced by The Jackson Laboratory’s Genome Technologies Core Facility. RNA-seq libraries were prepared with KAPA Stranded mRNA-Seq kit. Quantification of libraries was performed using real-time qPCR. Sequencing was performed on Illumina Hiseq 4000 platform (Illumina, San Diego, CA, USA), generating paired-end reads of 75 bp. Raw reads obtained from the sequencer were processed including quality control steps to identify and remove low-quality samples. Reads with more than 50% low-quality bases (>Q30) overall were filtered out and the remaining high-quality reads were then used for expression estimation. Alignment estimation of gene expression levels using the EM algorithm for paired-end read data was performed using RSEM (package version 1.2.19) with default settings. Bowtie2 was used as an aligner to align the mapped reads against the hg38 reference genome. Data quality control was performed using picard and bamtools to obtain general alignment statistics from the bam file. The expected counts data were normalized using the TMM normalization method.

### 2.10. Subtype-Specific Ion Channel Gene Expression

Ion channel genes were manually selected from the publicly available gene expression datasets of medulloblastoma (MB) subtypes [5], which showed enrichment in group 3 and 4 MBs compared to WNT and SHH subtypes. Ion channel gene expression analysis was performed using R (version 4.3.0) statistical software with rounded gene-level abundance estimates from RSEM. Gene annotations for transcripts were gathered from Ensembl (release 99) using biomaRt (version 2.42.0). Normalization of read counts, variance stabilizing transformation, filtering low gene variance, and detection of differential expression were performed using the DESeq2 (version 1.40.0) package (parameters: “alpha = 0.01”). Figures were created using the ggplot2 (version 3.4.2) and ComplexHeatmap (version 2.16.0) packages.

### 2.11. Statistics

All data were analyzed with one-way ANOVA test. Error bars indicate standard error of mean. IC50 was determined using Graphpad Prism software (version 9).

## 3. Results

### 3.1. Enriched Potassium Ion Channel Expression in MB Tissue and Patient-Derived Primary Cultures

In our laboratory, we have been developing patient-derived pediatric brain tumor models that closely mimic the genetic and phenotypic characteristics of primary tumor tissue [28]. Using publicly available gene expression datasets of medulloblastoma (MB) subtypes [5], we identified a significant enrichment of specific ion channel genes in group 3 and 4 MBs compared to the WNT and SHH subtypes.

To further investigate these findings, we analyzed RNA sequencing (RNAseq) data from pediatric brain tumor patient samples collected at Connecticut Children’s Medical Center between 2015 and 2017. The dataset included samples from medulloblastoma (*n* = 3), astrocytoma (*n* = 3), ependymoma (*n* = 3), ‘normal’ brain tissue resected during epilepsy surgery (*n* = 2), and patient-derived primary cultures (*n* = 44). Detailed RNA sequencing data and analysis of these patient samples and patient-derived cell cultures have been reported in our previous publications [28,29].

As shown in the heatmap in Figure 1, the MB tissue and the corresponding patient-derived cell cultures exhibited markedly elevated expression levels of several potassium channel genes, including KCNB2 (Kv2.2), KCNA5 (Kv1.5), KCNA1 (Kv1.1), and KCNJ2 (Kir2.1), when compared to normal brain tissue, as well as other brain tumor types and their respective primary cultures.

### 3.2. Subtype-Specific Potassium Channel Expression Pattern in Primary Cell Culture Derived from Patient with MB

To investigate the expression patterns of MB-associated ion channels, we performed immunostaining against mGluR8, Kv1.5, Kv2.2, Kir2.1, Kv1.1, Kv2.1, and sodium channels (Pan-NaV) (Figure 2A). The MB cells exhibited distinctive positive staining patterns depending on the ion channel type. For instance, mGluR8 was concentrated in cell aggregates (Figure 2A(a,b)). Kv1.5 expression was detected in specific cells along elongated processes (Figure 2A(c,d)). Kv2.2 was localized at sites where neurite-like processes originated (Figure 2A(e,f), white arrows). In contrast, the Kir2.1 and Pan-NaV channels were present in a large proportion of the cells without clear intracellular localization patterns.

Flow cytometry was used to quantify the percentages of cells expressing these specific channel proteins. As shown in Figure 2B, the histograms revealed cell populations with expression levels of 5.2% for mGluR8, 20.9% for Kv1.5, 9.1% for Kv2.2, 33.6% for Kir2.1, and 35.2% for Pan-NaV. Very few cells (<1%) expressed Kv2.1 or Kv1.1. 

### 3.3. Spontaneous Calcium Signals in MB DAOY Cells

In the DAOY medulloblastoma cells, calcium signal waves were observed as changes in the Fluo-4 fluorescence intensity (Figure 3). Fluo-4 live-stained DAOY cultures (Figure 3A) were recorded over a 1 min time lapse at 15 frames per second (fps). For analysis, 10 randomly selected cells within a 10× magnification field of view were used to measure the fluorescence signals (Figure 3B(a)).

To characterize the time-series signals, we developed an R script to detect significant peaks, defined as signals exceeding 3.5 times the baseline variance. The DAOY cells exhibited signal peaks with an average wavelength of approximately 40 *s* and a 2.5-fold amplification of their calcium signal intensity. Overall, 67 ± 23% of cells (n = 440) demonstrated spontaneous calcium signal peaks.

Table 1 shows the eight drugs we tested, including a calcium chelator (BAPTA-AM “BAP”), a calcium internal release inhibitor (ryanodine “RYAN”), a calcium channel blocker (mibefradil “MIB”), potassium channel blockers (paxilline “PAX”, thioridiazine “TOZ”), a Na/K pump inhibitor (digoxin “DIG”, ouabain “OUA”), and a sodium channel blocker (riluzole “RIZ”). This list includes FDA-approved drugs (MIB, TOZ, DIG, OUA, and RIZ) and tool compounds for ion channel activities (BAP, RYAN and PAX). bioengineering-12-00268-t001_Table 1Table 1Channel blockers and modulators.AbbreviationFull NameDrug ActionBAPBAPTA-AMCell-permeant Ca++ chelatorRYANRyanodineCalcium internal release blockerMIBMibefradil *Calcium channel blocker (T-type)PAXPaxillineA Blocker of KCa1.1 (BK) Channels and SERCA PumpsTOZThioridiazine *hERG potassium channel blockerDIGDigoxin *Na/K-ATPase inhibitorOUAOuabain *Na/K-ATPase inhibitorRIZRiluzole *Sodium channel blocker* FDA-approved.


To assess the effects of drug treatments, we tracked the same cells before and after drug administration (Figure 3B(b)). Drug treatments often resulted in the loss of spontaneous calcium signal peaks; however, neither the wavelength of the peaks nor the pulse frequency was altered by the drugs.Consequently, we quantified the percentage of cells displaying a calcium signal peak as the primary outcome measure and compared the changes within the same well before and after drug treatment.

To evaluate long-lasting drug effects, calcium signal changes were also analyzed at 20 and 40 min post-treatment. Since calcium signaling is highly sensitive, and repeated imaging of the same area can lead to fluorescence bleaching and false negatives, we used two separate sets of cultures: one for baseline and 1 min post-treatment imaging, and another for imaging at 20 and 40 min post-treatment.

### 3.4. Drug-Induced Calcium Signal Changes in DAOY Cells

Figure 3C illustrates the drug-induced changes in the calcium signals in the DAOY cells. Treatment with BAP (10 µM) completely blocked the Fluo-4 signal peaks in all cells at all observed time points (1 min, 20 min, and 40 min), confirming that the observed peaks were calcium signals. Conversely, treatment with RYAN did not suppress the internal calcium release, indicating that the calcium signals originated from external sources.

At 1 min post-drug treatment, BAP, MIB, PAX, TOZ, and DIG fully eliminated the Fluo-4 signal peaks in 100% of cells. In the cultures treated with OUA (10 µM), 12.5% of the cells still exhibited peaking activity at 1 min, which decreased to less than 10% at 20 and 40 min post-treatment. Overall, the cells treated with BAP, MIB, PAX, TOZ, DIG, and OUA retained less than 25% of their baseline activity levels (Figure 3C, dotted line). In contrast, the cultures treated with RIZ (20 µM) and RYAN (100 µM) maintained over 25% of their activity levels.

### 3.5. Cytotoxicity of Channel Blockers and Modulators in DAOY Cells

We assessed the cytotoxic effects of channel blockers and modulators in DAOY cultures using the Live/Dead viability assay. Figure 4 illustrates the dose–response curves, with the Live stain (calcein) fluorescence intensity normalized to control wells treated with 0.1% DMSO. Figure 4A–C depict the cytotoxicity profiles of the calcium channel modulators (BAP, MIB), potassium channel modulators (PAX, TOZ), and Na/K pump inhibitors (DIG, OUA).

At the highest doses tested (96 h treatment), BAP (50 µM), MIB (50 µM), PAX (50 µM), TOZ (50 µM), DIG (10 µM), and OUA (10 µM) achieved cell-killing rates of 85%, 85%, 95%, 94%, 96%, and 86%, respectively. Figure 4D highlights drugs with insignificant effects, including the sodium channel blocker RIZ, which showed a slight viability decrease to ~70% at 50 µM for 96 h, and the calcium store release inhibitor RYAN, which induced little change even at concentrations up to 100 µM.

### 3.6. Spontaneous Calcium Signals in Primary Cells of Patient with MB and Calcium Signal Blockade

We established non-passaged primary tumor cell cultures from the tumor tissue of one patient with MB (MB#51). The clinical diagnosis of this patient is group 3 or 4 MB (non-WNT, non-SHH, p53 wild-type). The cells were dissociated from the patient’s fresh tumor and cultured in a lab-optimized medium that preserves MB tumor molecular characteristics in vitro [29]. All the experiments were performed on cultures of cells directly dissociated from the fresh tumor tissue, or from cryopreserved tissue of the same tumor. None of the patient cells used in the study had been passaged prior to experimentation, as we previously found that cells had reduced drug sensitivity even after one passage. The primary cells exhibited a spindly morphology compared to the DAOY cells (Figure 5A).

These primary cells also displayed fluorescence signal peaks during 1 min recordings at 15 fps (Figure 5B). Among 100 randomly selected cells from 10 wells, an average of 84% (± 6%) demonstrated signal peaks. These peaks showed a two-fold magnitude change from the baseline with a duration of approximately 50 s.

Spontaneous calcium signals were found only in MB cells but not in the primary cultures derived from other brain tumor types (astrocytoma, ependymoma, glioblastoma), as reported in our previous studies [28,29].

To evaluate the calcium signal modulation, we tested channel blockers and modulators in day-7 primary cultures, focusing on 20 and 40 min post-drug treatment due to the limited cell quantities (Figure 5C). BAP (10 µM), MIB (50 µM), and PAX (50 µM) eliminated the Fluo-4 signal peaks in 100% of the cells at both time points. TOZ (20 µM) and RIZ (50 µM) reduced signal peaks in more than 50% of the cells.

In contrast, the DIG (10 µM)- and OUA (20 µM)-treated cultures retained Fluo-4 activity in over 50% of their cells at both time points. Interestingly, RYAN (100 µM) blocked Fluo-4 peaks in 90% of the cells at 20 min; however, 65% of cells regained peaking activity by 40 min.

### 3.7. Cytotoxicity of Channel Blockers and Modulators in Primary Cells of Patient with MB

We assessed the tumor cell-killing activity of MIB, PAX, TOZ, DIG, and OUA in MB#51 primary cell cultures using the Live/Dead viability assay (Figure 6). Due to the limited quantity of primary cells, BAP, RIZ, and RYAN were not tested on the MB#51 cells.

As shown in Figure 6A, the tumor cell-killing activity of MIB, PAX, and TOZ exhibited dose- and time-dependent decreases in viability. At the highest doses tested, MIB (100 µM), PAX (50 µM), and TOZ (100 µM) achieved cell-killing rates of 80%, 70%, and 85%, respectively, following 96 h of treatment.

In contrast, Figure 6B highlights that DIG and OUA demonstrated lower degrees of cytotoxicity. After 96 h of treatment at 10 µM, DIG and OUA achieved cell-killing rates of 47% and 52%, respectively, which further declined to 39% and 37% at 20 µM.

### 3.8. Comparison of Calcium Signal Suppression and MB Cytotoxicity by Channel Blockers and Modulators Between DAOY Cells and MB Primary Cells

We compared the cytotoxic effects of channel blockers and modulators on DAOY cells versus MB#51 primary cells (Figure 4 vs. Figure 6), the results of which are summarized as IC50 values in Table 2. MIB, PAX, and TOZ demonstrated effective dose ranges for both the DAOY and MB#51 cells. However, while DIG and OUA exhibited potent cytotoxicity at concentrations <1 µM in the DAOY cells, they lacked a clear dose-dependent effect in the MB#51 cells.

When analyzing the relationship between calcium signal suppression and cytotoxicity for all tested drugs (Table 3), an inverse correlation between post-treatment calcium activity and cell death was evident. In the DAOY cells, more than 80% of the cells lost calcium signaling within 40 min of treatment with MIB, PAX, TOZ, DIG, or OUA, which coincided with cell death exceeding 85%. In contrast, treatment with RIZ or RYAN resulted in only modest reductions in calcium signaling and cell viability.

In the MB#51 cells, MIB, PAX, and TOZ induced significant reductions in both calcium signaling and cell viability. Conversely, the DIG and OUA treatments did not substantially suppress calcium signaling or induce cell death.

### 3.9. Cytotoxicity of Channel Blockers and Modulators in D283 Cells

We evaluated the cytotoxic effects of channel blockers and modulators on another commonly used MB cell line, D283, following a similar approach to the tests conducted on DAOY cells. Unlike DAOY cells, D283 cells do not adhere to 2D culture plates, making calcium imaging impractical; as such, we did not systematically examine their calcium signaling profiles. Figure 7 presents the dose–response curves, with the Live stain (calcein) fluorescence intensity normalized to control wells treated with 0.1% DMSO. Figure 7A–C illustrate the cytotoxicity profiles of the calcium channel modulators (BAP, MIB), potassium channel modulators (PAX, TOZ), and Na/K pump inhibitors (DIG, OUA), demonstrating the positive dose-dependent killing of D283 cells. In contrast, Figure 7D shows the dose–response curves for the sodium channel blocker RIZ and the calcium store release inhibitor RYAN, indicating no significant cytotoxic responses to these drugs.

## 4. Discussion

Medulloblastoma (MB) groups 3 and 4 lack targeted therapies despite their poor patient outcomes. Ion channels and pumps are implicated as potential drivers of MB metastasis and growth, but the mechanistic connection between aberrant channel expression and tumor progression remains unclear. In this study, we repurposed FDA-approved ion channel blockers and modulators to investigate their anti-tumor potential in MB cells. For the first time, we observed spontaneous calcium signaling in MB cell line- and patient-derived primary cells. These signals were abolished by the calcium chelator BAPTA-AM but not by ryanodine, suggesting they arise from extracellular sources through ion channels in the plasma membrane. The calcium influx was significantly reduced by mibefradil, thioridazine, and paxilline in both the MB cell line and primary MB cells. Calcium signal inhibition was correlated with dose-dependent cytotoxicity induced by these drugs. While digoxin and ouabain, Na^+^/K^+^-pump inhibitors, suppressed calcium signaling and induced cytotoxicity in the MB cell lines, their effects on the primary MB cells were significantly less pronounced, highlighting potential differences in drug sensitivity between cell types. These findings establish calcium signaling as critical for MB cell growth and demonstrate that its inhibition using specific channel blockers can suppress tumor cell viability. This study further highlights the importance of patient-derived cells in drug testing, given the observed variability between MB cell lines and primary MB cells.

Our findings align with prior genetic studies that identified the enrichment of ion channel genes in MB groups 3 and 4 compared to other subtypes. Specifically, potassium channels such as Kv10.2 (EAG2), Kv2.2 (KCNB2), Kv1.5 (KCNA5), and Kir2.1 (KCNJ2) were upregulated in the MB tumor tissues and primary cultures. These genes are known to regulate cell proliferation, motility, differentiation, and calcium signaling, and the distinctive protein expression patterns demonstrated in this study provide further evidence of their involvement in MB cell growth. For instance, KCNB2 (Kv2.2) expression is linked to cellular extension at the junction of the cell body and the processes, correlating with cell mobility and differentiation [30]. KCNB2 is previously found to be enriched in MB-propagating cells that contribute to drug resistance and tumor relapse; and loss of the KCNB2 channel can deplete this population in part through increased differentiation [31]. The KCNA5 expression in long processes is consistent with its role in the differentiation and proliferation of human glia [32,33]. KCNA5 is also found to play an important role in regulating cell cycle transition and proliferation in a wide range of other cell types [10,34,35]. Kir2.1, encoded by KCNJ2, is suggested as a therapeutic target for non-WNT/non-SHH MBs [36]. The heterogeneity of potassium channel expression among primary cells reflects their diverse roles and emphasizes the complexity of calcium signaling regulation in MB.

Spontaneous calcium signaling, as we describe in this study, is rare in non-excitable cells and unique among MB cells when compared to other pediatric brain tumors such as glioblastoma, ependymoma, and astrocytoma. Repetitive Ca^2+^ oscillations have been implicated in the migration of glioma cells [37,38,39]. The slow-wave calcium signals observed in MB cells are distinct from the rapid calcium spikes seen in neurons, further indicating a tumor-specific mechanism. This study provides insights into the calcium-dependent pathways that drive MB growth, such as migration and cell proliferation [15,40,41].

Mibefradil was found to block calcium influx and induce cytotoxicity in both DAOY and MB primary cells. The inhibition of cell proliferation by mibefradil has been demonstrated in human U87 MG glioma and murine neuroblastoma cell lines [42], as well as in glioblastoma xenografts and tumor stem cells [15,43,44]. These preclinical studies had led to clinical trials using mibefradil to sensitize glioblastoma to chemotherapy [21]. The observed effect of mibefradil in MB groups 3 and 4 warrants the further exploration of T-type calcium channels in tumorigenesis and progression.

Thioridazine’s suppression of calcium signals and induction of cell death in MB cells offers a mechanistic link to its inhibition of potassium channels, particularly hERG (Kv11.1) channels. The aberrant expression of hERG channels is found in many cancer types [45], and thioridzine inhibition is found to induce cell death in a wide range of cancers including cervical, prostate, gastric, B cell lymphoma, and lung cancer, as well as in neuroblastoma [46,47,48,49,50,51]. Our findings add to prior research demonstrating thioridazine’s anti-cancer activity through various mechanisms, including PI3K/Akt/mTOR pathway suppression [46] and as a dopamine receptor antagonist [52]. Additionally, thioridazine has been shown to inhibit EAG2 channels, preventing the cell volume changes required for tumor cell migration and invasion [6,53]. Calcium signaling also plays a key role in cell volume regulation [54]. Therefore, a link between calcium signal suppression and MB cell death caused by thioridazine may further elucidate the role of the hERG channel in MB growth.

Paxilline’s potent inhibition of calcium signaling and its cytotoxicity in MB cells indicate a critical role for the large-conductance potassium (BK) channel in tumorigenesis. The BK channel belongs to the group of Ca^2+^-activated potassium channels (KCa) that couple the increase in intracellular Ca^2+^ concentration to hyperpolarization of the membrane potential [55]. These KCa channels can also affect the driving force for Ca^2+^ influx and thus exert a secondary control of intracellular calcium levels. BK channels have previously been implicated in GBM cell lines [56], further suggesting their relevance in MB.

As cardiac glycosides and potent inhibitors of Na^+^/K^+^-ATPase [57], digoxin and ouabain were first identified to have anti-MB activity in Huang’s large-scale drug screening study [7]. The mechanism of digoxin’s anti-proliferative effects in MB remains unclear. However, studies in other cancers suggest roles beyond Na^+^/K^+^-ATPase inhibition, such as proteasome activation and the modulation of estrogen receptor α expression in breast cancers [58]. Notably, digoxin and ouabain have been reported to increase the intracellular calcium levels in cardiac myocytes [57]. Despite the differential effects of digoxin on the DAOY and D283 cell lines and primary MB cells, there is a consistent correlation between calcium signaling inhibition and cytotoxic activity. In primary cells, the relatively smaller degree of cell death was accompanied by residual calcium signaling activity, whereas the treated cell lines exhibited the complete loss of both calcium signaling and viability.

Together, these findings demonstrate that spontaneous calcium signaling plays a critical role in MB progression and that ion channel blockers can effectively induce MB cell death. While evaluating the effects of these drugs on normal brain cells would provide valuable insights, obtaining such cells for in vitro studies remains a significant challenge. Although we previously attempted to culture normal human brain cells from patient samples [59], these efforts have not yet advanced to a stage that permits drug testing. Induced pluripotent stem cell-derived brain cell or tissue models could serve as valuable companion models to our patient-derived tumor cultures for future drug screening [60]. Nevertheless, several lines of evidence in our study support the tumor-specific effects of these ion channel inhibitors. Specifically, the MB-specific ion channel gene enrichment and the presence of spontaneous calcium signaling in MB—contrasting with its absence in other brain tumor types—suggest that the observed effects are not merely due to general cytotoxicity.

Despite these promising findings, several questions remain. First, potential off-target effects and drug specificity limitations may complicate the interpretation of our results. For example, thioridazine, a dopamine receptor antagonist, may induce apoptosis through interactions with dopamine receptors on MB cells [61] rather than solely through ion channel modulation [62,63]. Second, the similar drug sensitivities observed in the two MB cell lines, DAOY (SHH subtype) and D283 (group 3/4 subtype), raise questions about the specificity of these treatments across MB subtypes. While the group 3 or 4 MB patient-derived primary cells also responded to the drugs, the discrepancy in sensitivity to digitalis drugs between the MB cell lines and patient-derived cells highlights the complexity of molecular subtypes, emphasizing the need for further studies using a broader range of patient samples.

Additionally, while repurposing FDA-approved drugs for MB based on shared molecular targets offers cost-effective and accelerated therapeutic development, challenges remain. Key pharmacokinetic factors—including blood–brain barrier penetration, systemic toxicity (as seen with digoxin [64]), and drug–drug interactions affecting other organ systems (e.g., digoxin for cardiac function and mibefradil for liver metabolism [65])—must be carefully considered. Nevertheless, this study establishes a foundation for targeting ion channels in high-risk MB subtypes. Future investigations using advanced methodologies such as electrophysiology, gene editing, and RNA interference will be crucial in further elucidating the functional roles of specific ion channels in MB pathogenesis.

## 5. Conclusions

For the first time, we report spontaneous calcium signaling in MB cells. This finding underscores the importance of calcium signaling in MB tumor progression and provides a foundation for the therapeutic targeting of MB. Among the channel blockers and modulators that we tested, mibefradil, thioridazine, and paxilline effectively inhibited calcium signaling and induced cytotoxicity in MB cells. Digoxin and ouabain showed similar effects on MB cell lines but were less potent in primary MB cells.

Our results highlight calcium signaling as a key mechanism underlying MB cell growth and validate ion channel blockers as potential candidates for new therapies that target MB groups 3 and 4. The observed differences between the MB cell lines and patient-derived cells emphasize the importance of using diverse patient samples in future studies to capture the heterogeneity of MB. This study identifies ion channel-mediated calcium signaling as a promising druggable target to address the urgent need for effective therapies for high-risk MB groups.

## Figures and Tables

**Figure 1 bioengineering-12-00268-f001:**
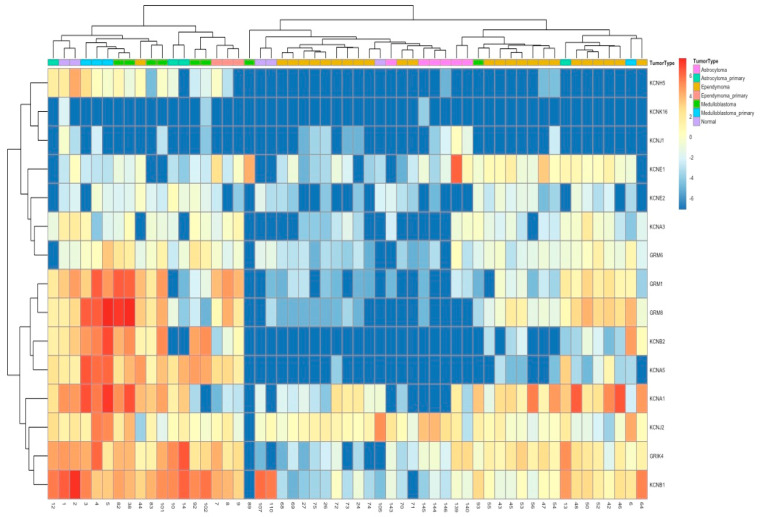
Relative ion channel gene expression in MB primary tumors and cultures. The heatmap illustrates the relative expression levels of ion channel genes identified in the literature as dysregulated in MB. Red denotes higher levels of gene expression, and blue represents lower levels. Tumor types are indicated along the top x-axis, with blue labels corresponding to primary medulloblastoma (MB) tumors and green labels representing cultures derived from primary MB tissue. Both primary MB tumors and their derived cultures exhibit upregulated expression of potassium channel genes, including KCNB2, KCNA5, KCNA1, KCNJ2, GRK4, and KCNB1 (along the right y-axis). In contrast, astrocytoma, ependymoma, and their associated cultures show relatively lower expression levels of these ion channels.

**Figure 2 bioengineering-12-00268-f002:**
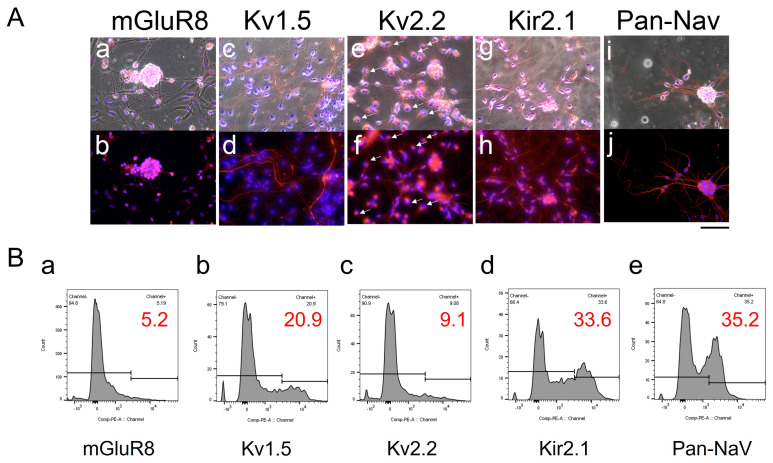
**Ion channel expression in medulloblastoma (MB) primary culture.** (**A**). Representative immunostaining images showing the expression of channel proteins, including **mGluR8** (a,b), **Kv1.5** (c,d), **Kv2.2** (e,f), **Kir2.1** (g,h), and **Pan-NaV** (i,j). Channel proteins are visualized in red and counterstained with DAPI (blue) for nuclear staining. Overlays of phase-contrast images (a,c,e,g,i) demonstrate the spindly morphology of MB cells in culture. White arrows in e and f indicate Kv2.2 localization at sites where neurite-like processes originate. Scale bar, 100 um. (**B**). Flow cytometry analysis histograms of cells derived from a patient with MB depicting positive expression for specific channel proteins: mGluR8 (a), Kv1.5 (b), Kv2.2 (c), Kir2.1 (d) and Pan-NaV (e). Percentages indicate the proportion of cells in the positive gate, with gating thresholds established using secondary antibody-only controls (no primary antibody staining).

**Figure 3 bioengineering-12-00268-f003:**
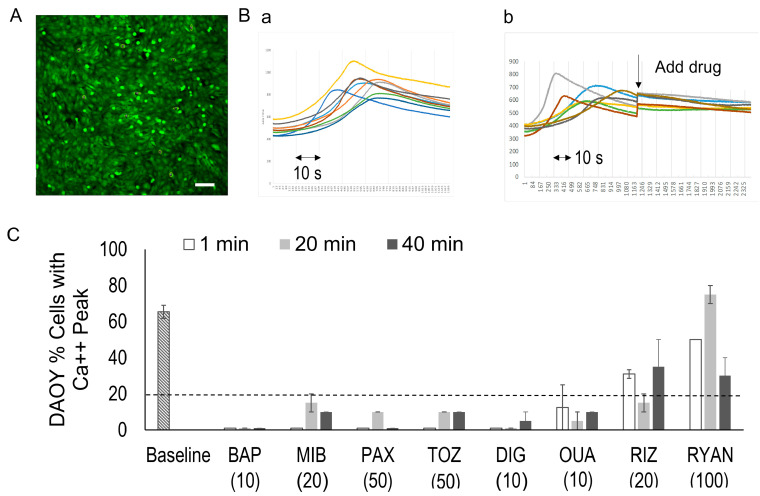
**Spontaneous calcium signals in DAOY cells and drug-induced signal changes.** (**A**). Representative fluorescence image of DAOY cells loaded with the calcium indicator Fluo-4. Scale bar, 100 µm. (**B**). Fluorescence signal time-series of randomly selected DAOY cells for 1–2 min recording at 15 frames per second (fps). (a) Baseline signal during 1 min recording. (b) Signal tracking of the same cells before and 1 min post-drug administration. (**C**). Percentages of cells exhibiting calcium signal peaks in untreated cultures (baseline) and after drug treatment. Drug abbreviations are provided in Table 1. Numbers in parentheses indicate drug doses (in µM). The dotted line represents 25% of the baseline value.

**Figure 4 bioengineering-12-00268-f004:**
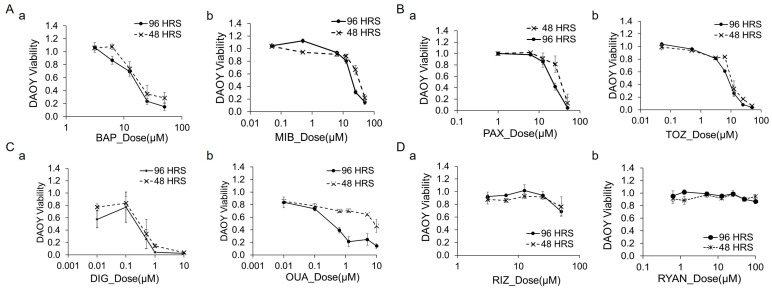
**Cytotoxicity of channel blockers and modulators in DAOY cells.** Dose–response curves of DAOY cells treated with various drugs. The y-axis represents cell viability measured by calcein fluorescence intensity using the Live/Dead assay, normalized to control wells treated with 0.1% DMSO. The x-axis shows drug concentrations on a logarithmic scale. Each data point represents the mean of five replicates. (**A**). Calcium channel modulators: (a) BAP and (b) MIB. (**B**). Potassium channel modulators: (a) PAX and (b) TOZ. (**C**). Na/K pump inhibitors: (a) DIG and (b) OUA. (**D**). Additional modulators: (a) sodium channel blocker RIZ and (b) intracellular calcium store-release inhibitor RYAN.

**Figure 5 bioengineering-12-00268-f005:**
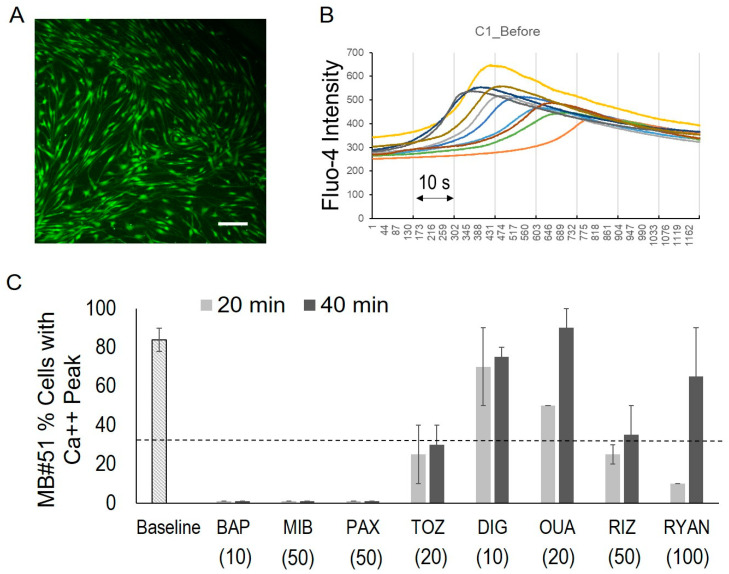
**Spontaneous calcium signals in patient with MB (#51) primary cells and drug-induced signal changes.** (**A**). Representative fluorescence image of MB#51 cells in culture, 7 days post-dissociation from fresh patient tumor tissue. Cells were loaded with the calcium indicator Fluo-4. Scale bar, 100 µm. (**B**). Fluorescence signal time-series of randomly selected MB#51 cells recorded over 1 min at 15 frames per second (fps). (**C**). Percentage of cells exhibiting calcium signal peaks in untreated cultures (baseline) and following drug treatment. Drug abbreviations are listed in Table 1, with concentrations provided in parentheses (µM). The dotted line marks 25% of the baseline value.

**Figure 6 bioengineering-12-00268-f006:**
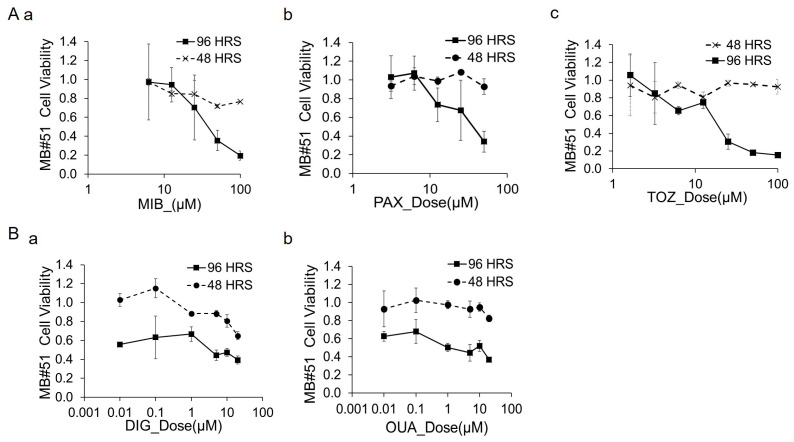
**Cytotoxicity of channel blockers and modulators in patient with MB (#51) primary cells.** Dose–response curves of 7-day-cultured MB#51 cells treated with various drugs. The y-axis represents cell viability, measured by calcein fluorescence intensity using the Live/Dead assay and normalized to control wells treated with 0.1% DMSO. The x-axis shows drug concentrations on a logarithmic scale. Each data point represents the mean of five replicates. (**A**). (a) Calcium channel modulator MIB. (b) Potassium channel modulator PAX. (c) Potassium channel modulator TOZ. (**B**). Na/K pump inhibitors: (a) DIG and (b) OUA.

**Figure 7 bioengineering-12-00268-f007:**
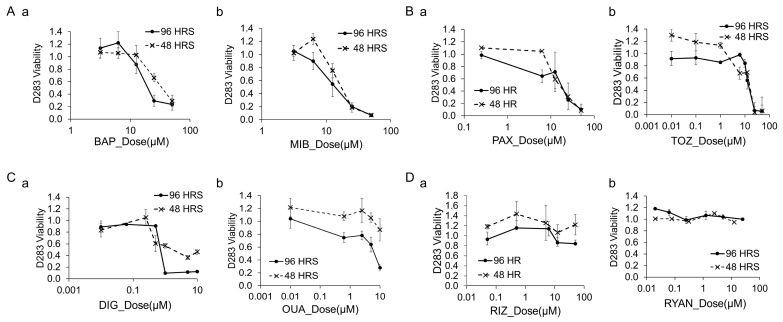
**Cytotoxicity of channel blockers and modulators in D283 cells.** Dose–response curves of D283 cells treated with various drugs. The y-axis represents cell viability measured by calcein fluorescence intensity using the Live/Dead assay, normalized to control wells treated with 0.1% DMSO. The x-axis shows drug concentrations on a logarithmic scale. Each data point represents the mean of five replicates. (**A**) Calcium channel modulators: (a) BAP and (b) MIB. (**B**) Potassium channel modulators: (a) PAX and (b) TOZ. (**C**) Na/K pump inhibitors: (a) DIG and (b) OUA. (**D**) Additional modulators: (a) sodium channel blocker RIZ and (b) intracellular calcium store-release inhibitor RYAN.

**Table 2 bioengineering-12-00268-t002:** IC50 (µM) of channel blockers and modulators.

Abbrevation	Full Name	DAOY—96 h	MB#51—96 h
MIB	Mibefradil	16.66	31.90
PAX	Paxilline	23.22	19.61
TOZ	Thioridiazine	7.86	17.88
DIG	Digoxin	0.44	n/a
OUA	Ouabain	0.27	n/a

**Table 3 bioengineering-12-00268-t003:** Calcium imaging and cytotoxicity comparison.

Abbrevation	Full Name	DAOY		MB#51	
		% with Ca++ Peak After 40 Min	Maximum Cell-Kill (%)	% with Ca++ Peak After 40 Min	Maximum Cell-Kill (%)
BAP	BAPTA-AM	0 (10 µM)	85 (50 µM)	0 (10 µM)	not tested
MIB	Mibefradil	10 (20 µM)	85 (50 µM)	0 (50 µM)	80 (100 µM)
PAX	Paxilline	0 (50 µM)	95 (50 µM)	0 (50 µM)	66 (50 µM)
TOZ	Thioridiazine	10 (50 µM)	96 (50 µM)	30 (20 µM)	85 (100 µM)
DIG	Digoxin	5 (10 µM)	97 (10 µM)	75 (10 µM)	61 (20 µM)
OUA	Ouabain	18 (10 µM)	86 (10 µM)	90 (20 µM)	63 (20 µM)
RIZ	Riluzole	35 (20 µM)	31 (20 µM)	35 (50 µM)	not tested
RYAN	Ryanodine	30 (100 µM)	13 (100 µM)	65 (200 µM)	not tested

## Data Availability

The RNAseq data presented in this study can be accessed from the Sequence Read Archive (SRA) on NCBI website (SUB10713222). Tissue samples can be accessed upon request to Dr. Ching Lau (Ching.Lau@jax.org) at the Jackson Laboratory.

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
