# Peer review of "Targeting Ion Channels: Blockers Suppress Calcium Signals and Induce Cytotoxicity Across Medulloblastoma Cell Models"

_bioengineering, 2025, doi:10.3390/bioengineering12030268_

Round 1

Reviewer 1 Report

Comments and Suggestions for Authors

The authors presented a well-structured study with new ideas of the exploring the potential of repurposing FDA-approved ion channel  blockers as treatments for groups 3 and 4 medulloblastoma.

Recommendations for Revision and Improvement

The introduction requires more depth and should present a stronger rationale for the study. Expanding on the significance of targeting ion channels for cancer therapy (Their function is required for many characteristic features of the aggressive cancer cell behavior such as migration, invasion and metastasis as well as proliferation and therapy resistance). These potential applications would strengthen the context.

In the first paragraph, it is recommended to cite additional relevant references or reviews to support the claims and provide a comprehensive background.

This review would probably be helpful:

Xiaohua Liu, Chunyong Ding, Wenfu Tan, A. Zhang,  Medulloblastoma: Molecular understanding, treatment evolution, and new developments. 2020, 210,  107516.

By addressing these points, the study will achieve a higher level of clarity, further supporting its publication.

Author Response

The introduction requires more depth and should present a stronger rationale for the study. Expanding on the significance of targeting ion channels for cancer therapy (Their function is required for many characteristic features of the aggressive cancer cell behavior such as migration, invasion and metastasis as well as proliferation and therapy resistance). These potential applications would strengthen the context.

Reply: We thank the reviewer for this suggestion. We have added a new paragraph in Intro. on the roles of ion channels in cancer cell biology.

In the first paragraph, it is recommended to cite additional relevant references or reviews to support the claims and provide a comprehensive background.

This review would probably be helpful:

Xiaohua Liu, Chunyong Ding, Wenfu Tan, A. Zhang,  Medulloblastoma: Molecular understanding, treatment evolution, and new developments. 2020, 210,  107516.

Reply: We provided this additional reference in the revised manuscript (no. 4).

Reviewer 2 Report

Comments and Suggestions for Authors

The experimental design is comprehensive and appropriate for the research objectives

Author Response

The experimental design is comprehensive and appropriate for the research objectives

Reply: We thank the reviewer for your positive endorsement of our manuscript.

Reviewer 3 Report

Comments and Suggestions for Authors

In the study, the potential of ion channel blockers for the treatment of groups 3 and 4 Medulloblastoma was investigated. The study is novel and interesting, however there are some issues which should be enlightened.

-            It was mentioned that the drugs were used to treat groups 3 and 4 Medulloblastoma but there is no information about which cells correspond these specific groups. Detailed information about subgroups of cell line and tissues should be given.

-            3.2. Subtype-specific Potassium Channel Expression Pattern in MB Patient-derived Primary Cell  Culture Section: A flow cytometry analysis results were given in this section but there is no information about this study in method section. Detailed information should be given in method section.

-            Detailed information about Subtype specific Potassium Ion Channel Expression analysis should be given in method section.

Author Response

It was mentioned that the drugs were used to treat groups 3 and 4 Medulloblastoma but there is no information about which cells correspond these specific groups. Detailed information about subgroups of cell line and tissues should be given.

Reply: We thank the reviewer for the suggestion. We clarified the molecular subtypes in the revised Introduction – “Experiments were conducted on MB cell lines (DAOY25 and D28326) as well as primary cell cultures of a MB patient. While the MB cell lines were established decades ago, lat-er molecular profiling classified DAOY as SHH type and D283 as group 3 or 4 27. The primary MB cells used for the live imaging and drug studies were derived from group 3 or 4 MB (non-WNT, non-SHH, p53 wild type).”

-            3.2. Subtype-specific Potassium Channel Expression Pattern in MB Patient-derived Primary Cell  Culture Section: A flow cytometry analysis results were given in this section but there is no information about this study in method section. Detailed information should be given in method section.

Reply: We thank the reviewer for pointing out this mishap. We carefully re-worked the manuscript and provided missing methods in the revised version, with new sections of “2.7. Flow Cytometry for Ion Channel Expression” and “2.8. Immunofluorescence Staining and Imaging’

-            Detailed information about Subtype specific Potassium Ion Channel Expression analysis should be given in method section.

Reply: We thank the reviewer for the suggestion. We provided detailed method in the new sections of “2.10. Subtype-specific Ion Channel Gene Expression”.

Reviewer 4 Report

Comments and Suggestions for Authors

The authors have demonstrated a promising exploration of ion channel blockers and modulators for medulloblastoma (MB) treatment. However, there are several potential drawbacks and challenges to be considered:

1.       The ion channels inhibitors used in the study may pose off target effects and lack drug specificity, like Thioridazine targets hERG potassium channels but also affects dopamine receptors, leading to potential neurological side effects. The therapeutic effects may not be from the ion channel inhibition causing difficulty in interpretating the data. To address this, Authors should try to report any off-target effects seen for the inhibitors used in the experiment to validate the study’s conclusions.

2.       As this study focuses on a brain tumor model, many of the ion channel inhibitors are known to have poor blood brain barrier permeability. Did the authors evaluate the permeability of these  drugs or measured the concentration of drug in the brain ?

3.       Systemic toxicity of the usage of cardiac glycosides, Digoxin’s narrow therapeutic index and systemic toxicity could limit the use as an anti-MB therapy. These effects could be particularly concerning in pediatric population. Mibefradil is withdrawn from the market due to serious drug drug interactions. Can the authors mention specific rationale for why this drug was included in the study?

4.       Did the authors try to evaluate the effects of these ion channel inhibitors on normal cell lines and compare the effects, this would help whether the effects are tumor specific or due to cytotoxicity. If done, please report the relevant data it would enhance the clarity and readability.

Author Response

.       The ion channels inhibitors used in the study may pose off target effects and lack drug specificity, like Thioridazine targets hERG potassium channels but also affects dopamine receptors, leading to potential neurological side effects. The therapeutic effects may not be from the ion channel inhibition causing difficulty in interpretating the data. To address this, Authors should try to report any off-target effects seen for the inhibitors used in the experiment to validate the study’s conclusions.

Reply: We thank the reviewer for the suggestion. We did not examine off-target effects of the drugs we used, as they are out of the scope of the study. We provided a separate section on this topic in the revised Discussion (line 547-556).

  1. As this study focuses on a brain tumor model, many of the ion channel inhibitors are known to have poor blood brain barrier permeability. Did the authors evaluate the permeability of these  drugs or measured the concentration of drug in the brain ?

Reply: We thank the reviewer for the suggestion. The study is on in vitro cell culture models, and given that these drugs have been previously approved by FDA, so we didn’t need to evaluate drug permeability. We agree that for further therapy development, drug permeability in the brain must be addressed, and we provided a sperate section on this topic in the revised Discussion (line 557-562).

  1. Systemic toxicity of the usage of cardiac glycosides, Digoxin’s narrow therapeutic index and systemic toxicity could limit the use as an anti-MB therapy. These effects could be particularly concerning in pediatric population. Mibefradil is withdrawn from the market due to serious drug drug interactions. Can the authors mention specific rationale for why this drug was included in the study?

Reply: We thank the reviewer for pointing out the limitations of these drugs. The study is focused on the mechanistic link between calcium signaling and medulloblastoma cell killing; therefore, these ion channel blockers are primarily used as tools to manipulate cell behavior. Because these drugs have been well characterized due to their FDA-approval status, it allows us to focus on the drug effects without worrying about unknown actions from un-characterized chemistry of other compounds. We acknowledge the limitations of these drugs in the revised Discussion (line 557-561).

  1. Did the authors try to evaluate the effects of these ion channel inhibitors on normal cell lines and compare the effects, this would help whether the effects are tumor specific or due to cytotoxicity. If done, please report the relevant data it would enhance the clarity and readability.

Reply: We thank the reviewer for the suggestion. Normal human brain cells are unfeasible to obtain for in vitro studies; we were the first to attempt to grow normal brain cells from human patient, but have not yet reached a point to allow for drug testing. The MB specific effects are implicated by other tumor type-specific findings, such as MB-specific ion channel gene enrichment, and calcium signals in MB but not other brain tumor types. We further highlighted these points in the manuscript (line 545-556).

Round 2

Reviewer 3 Report

Comments and Suggestions for Authors

The manuscript can be accepted in its current form.

Author Response

We thank the reviewer for positive endorsement of the revised manuscript.